# Limited Dorsal Myeloschisis with and without Type I Split Cord Malformation: Report of 3 Cases and Surgical Nuances

**DOI:** 10.3390/medicina55020028

**Published:** 2019-01-27

**Authors:** Yusuf Izci, Cahit Kural

**Affiliations:** Department of Neurosurgery, University of Health Sciences, Gulhane Education and Research Hospital, 06010 Ankara, Turkey; cahitkural23@gmail.com

**Keywords:** limited dorsal myeloschisis, split cord malformation, newborn, surgery

## Abstract

Limited dorsal myeloschisis (LDM) is a rare form of spina bifida which is characterized by a fibroneural stalk between the inner part of the skin and the spinal cord. It may be associated with split cord malformation (SCM). Diagnosis and management of this complex malformation is challenging. We presented 3 different cases of LDM. Two of them were associated with Type I SCM and the other had no associated malformation. All of them were evaluated radiologically just after the birth and underwent surgical treatment under intraoperative neuromonitoring. They were discharged without any complication. Newborns with spinal cystic lesions should be carefully evaluated for spinal malformations after the birth and treated surgically as soon as possible in order to prevent neurological and urological complications secondary to tethered cord syndrome. Surgical technique in LDM-SCM patients is quite different than the patients with solitary LDM.

## 1. Introduction

Limited dorsal myeloschisis (LDM) is a rare form of open neural tube defects, without an overtly unfused and exposed neural plate [1]. The constant features of an LDM are: A focal, closed, midline defect and a fibroneural stalk lying between the skin lesion and the spinal cord [1,2].

LDM may be seen in any part of the spinal neuraxis, but frequently observed in the lumbar region. Cervical LDM is very rare [1]. A skin-based, cerebrospinal fluid (CSF)-filled sac with a dome of tissues other than full-thickness skin is the common physical finding of LDM. However, LDM may also present as a non-saccular (flat) lesion [1,2,3]. Frequent associated pathologies are hydrocephalusand split cord malformation (SCM) [2,3]. The association of LDM with SCM is probably not coincidental but has important implications in their genesis [1,3]. Fibroneural stalks of LDM-SCM may be the persistent dorsal remnant of an anomalous ecto-endodermal fistula formed during early gastrulation [4,5]. The dorsal myeloschisis is restricted to the dorsal columns of the spinal cord. The long motor tracts and anterior columns are not involved [1,6].

Since the first description of LDM, a few reports have been published on the coexistence of LDM and SCM [1,6,7]. None of them had type I SCM and no detailed surgical technique was described.

In this report, we presented 3 newborns with cervical and lumbar LDMs. Two of them had Type I SCM below the fibroneural stalk level and the other was a solitary lesion without an associated malformation. Radiological and surgical features of these cases are documented and surgical nuances are emphasized.

## 2. Case Reports and Surgical Techniques

### 2.1. Case 1:

A 3-day old male newborn presented with a cystic mass lesion in his neck. He had no neurological deficit. The magnetic resonance imaging (MRI) revealed a cystic sac filled with CSF and a stalk lying from the cervical spinal cord to the inside of the sac covered with a thick skin. The stalk leaved the spinal cord at C3-C4 level (Figure 1A–C). The diagnosis was cervical LDM. He underwent surgical treatment under intraoperative neuromonitoring (Figure 1D). The dura was opened on the midline to expose the fibroneural stalk which was attached to the spinal cord. It was cut, and the spinal cord was released after the resection of meningocele sac (Figure 1E,F). No electrophysiological deterioration occurred after the removal of fibroneural stalk. The dura and the skin were closed and the patient was discharged without neurological deficit.

### 2.2. Case 2:

A 4-day old child presented with a cervical cystic lesion in his neck. He had no neurological deficit. His MRI revealed a stalk lying from the C3-C4 spinal cord to the inner part of the meningocele sac. There was also a septum below the stalk level (Figure 2A–C). The computed tomography (CT) showed that the septum was a bony spur with some fragmentations, probably not ossified (Figure 2D). The diagnosis was cervical LDM associated with Type I SCM (Figure 2E,F). He underwent surgical treatment under intraoperative neuromonitoring. Firstly, the meningocele sac was dissected and the distal part of the stalk was cut, and the meningocele sac was removed. Then, the bony septum was found, dissected from the dural sleeve and removed in piecemeal fashion under microscope (Figure 2G). The dura mater was incised and the dural sleeve was removed. The fibroneural stalk was originated just above the hemicords. The stalk was cut just on its attachment point to the spinal cord. The proximal spinal cord and hemicords were released, and the dura mater was closed (Figure 2H,I). No electrophysiological disturbance occurred during the surgery. The patient was discharged without neurological deficit (Figure 2J).

### 2.3. Case 3:

A 12-day old female patient was presented with a cystic mass lesion in her thoracolumbar region (Figure 3A). She had no neurological deficit. Her MRI showed a stalk leaving the spinal cord at L2 level and attached to the inner part of meningocele sac. There was also a septum dividing the spinal cord below the stalk (Figure 3B–D). The septum was a bony spur (Figure 3E). There was also syrinx cavity just above the stalk and at the cervical spinal cord. Other parts of neuraxis were normal. She underwent surgical treatment under intraoperative neuromonitoring. The meningocele sac was opened and the stalk was dissected from the inner wall of meningocele sac. Then, the bony septum was dissected from the dural sleeve and removed in piecemeal fashion (Figure 3F). The dura mater was opened and the dural sleeve was removed. It was observed that the stalk was originated just above the hemicords. The stalk was cut and lumbar spinal cord was released (Figure 3G). The fibrous bands around the hemicords were also cut and removed. The proximal spinal cord and the hemicords were released. Then, the dura mater was closed (Figure 3H). No electrophysiological deterioration was observed during the surgery. The patient was discharged without neurological deficit.

## 3. Discussion

The classification and nomenclature of cystic spinal lesions is controversial and various authors have suggested different classifications such as spina bifida cystica, cystic myelomeningocele, myelocystocele, and meningocele [8,9,10,11,12,13,14]. LDM was first described in 1993, and the lack of primitive neural placode differs LDM from the myelomeningocele [2,6,15].

SCM is a rare form of spinal dysraphism which is characterized by division of the spinal cord in two hemicords with a bony or fibrous septum [4,16,17,18]. There are two types of SCM. In type I, each hemicord is covered with its own dura mater and the hemicords are separated by a bony structure. In type II, the two hemicords are covered by a single thecal sac and separated by a fibrous tissue. Skin lesions such as hypertrichosis, dimple, capillary hemangioma, and nevus are typical findings of SCM [17,19]. Surgery of the type I SCM is unique like no other spinal surgery and has its own technique. Removal of the bony septum, opening the dura mater, resection of any other local spinal cord attachments causing tethering, excision of the dural sleeve, and dural reconstruction is the classical technique for type I SCM [18]. Although it is very rare, LDM and Type I SCM may simultaneously occur in the same patient. If the adequate radiological examinations were not performed, these malformations may be overlooked, and tethered cord syndrome may develop in childhood or adulthood.

MRI is the gold standard for the diagnosis of LDM and SCM [4,5,16,17]. The imaging hallmark of LDM is the visualization of the stalk that links the skin or posterior mass to the underlying spinal cord [2,3,6,15]. CT is useful to determine the nature of septum in SCM [16]. This septum may be connected with the vertebra body or dorsally situated without connection to the vertebra body [18]. In addition, the bony septum may be fragmented or non-ossified in newborns. This information may facilitate the surgical removal of bony septum in Type I SCM [18]. In the previous series of LDM-SCM association, splitting lesion was reported as below the fibroneural stalk or in the other parts of the spinal cord. Recently, Morioka et al. [6] presented 4 cases of LDM and one of them had Type II SCM. They performed histological analysis of the stalks and concluded that the diagnosis of LDM should be made based on comprehensive examination of histologic findings, as well as clinical manifestations [6]. Singh et al. [20] reported 12 cases of cystic cervical dysraphism and 4 of them had SCM. One of 4 cases had type I SCM at thoracic level and 3 cases had type II SCM at cervical level [20].

There is no controversy on the surgical treatment of LDM. The primary goal is the elimination of tethering on the spinal cord [2,5,6]. This surgery consisted of resection of the sac, cutting the stalk, and dissection and resection of other fibrous or neural structures that are connected to the dorsal side of dural tissue, which is finally closed [5]. However, the principal stage of the surgical treatment is the resection of stalk, regardless of whether the stalk contains neural nodules, complex peripheral nerves, or large vessels [2,5,6]. The stalk may be cut a few millimeters away from the spinal cord or just on the attachment point to the spinal cord. This may be checked by intraoperative stimulation of the stalk, proximal and distal spinal cords. Saccular lesions in the cervical or lumbar regions may be erroneously treated with ligation and cosmetic restoration of the large meningocele sac [1]. This may cause neurological deterioration in the future. All adjacent dysraphic lesions are also simultaneously treated, especially SCMs and spinal lipomas. However, in the presence of SCM below the stalk, the bony septum should be carefully dissected before the dural opening and then removed using microsurgical techniques. The stalk should be cut after the removal of the bony septum.

Pang et al. [15] reported 9 patients with cervical myelomeningocele in 1993, and there were only 2 patients with Type II SCM as associated malformation. No Type I SCM was reported [15]. In the cases described by Pang et al. [1], Salomao et al. [21], Habibi et al. [10], and Andronikon et al. [8], the neural stalks were almost contiguous with the median fibrous septum of the type II SCM, and the hemicords were directly adjacent to the part of the cord bearing the dorsal myeloschisis. However, in our 2 cases of LDM with Type I SCM, the stalk was not contiguous with the bony septum (or the dural sleeve) of the type I SCM, but the hemicords were directly adjacent to the part of the spinal cord bearing the dorsal myeloschisis.

Since LDM and SCM are tethering lesions, all of the published papers agree on the early surgical intervention of these entities, definitely before the onset of neurological deficits [1,2,5,6]. Pang et al. [1] reported a series of 51 patients with LDM and Type II SCM adjacent to LDM was seen, as an associated lesion, in 7 cases. They resected the fibrous septum of adjacent or non-adjacent SCMs to eliminate their independent tethering effect [1]. Lee et al. reported 33 patients with LDM and none of them had SCM associated with LDM [2]. Shashank et al. [7] reported a case of LDM associated with diplomyelia with dorsal bony spur, sacral meningocele, and syringohydromyelia. This case was a newborn and they performed early surgical treatment for each lesion [7]. Huang et al. [11] reported 10 patients with cervical myelomeningocele in 2010 and only one had Type I SCM as associated malformation. However, they did not give any information about the location and treatment of SCM associated with myelomeningocele [11]. Kasliwal et al. [22] reported 10 children with cervical myelomeningocele in 2007 and only one patient had type II SCM. In this case, the level of cervical myelomeningocele was C3-C4, and the SCM was located at C6 level [22]. In our cervical LDM-Type I SCM case, the bony septum was at C4 level just below the meningocele sac. The bony septum was large and irregular. Ossification of the septum was not completed yet. We resected the septum, cut the stalk which was attached to the meningocele sac, and detethered the spinal cord. Moreover, the hemicords were stimulated with bipolar stimulator to confirm the neurophysiological activity of both hemicords below the splitting level. In our lumbar LDM-Type I SCM case, the bony septum was also below the stalk level. We performed the same technique and detethered the spinal cord.

## 4. Conclusions

Surgical technique for the treatment of LDM associated with Type I SCM is quite different than the solitary LDM. Removal of the bony septum before cutting the fibroneural stalk is important for the adequate release of the spinal cord. This operation should be performed as soon as possible, before the ossification of the septum, in order to prevent neurological or urological deterioration. Intraoperative neuromonitoring is useful to check the functionality of spinal cord, hemicords, and aberrant rootlets that penetrate into the meningocele sac and provides safe surgery for children.

## Figures and Tables

**Figure 1 medicina-55-00028-f001:**
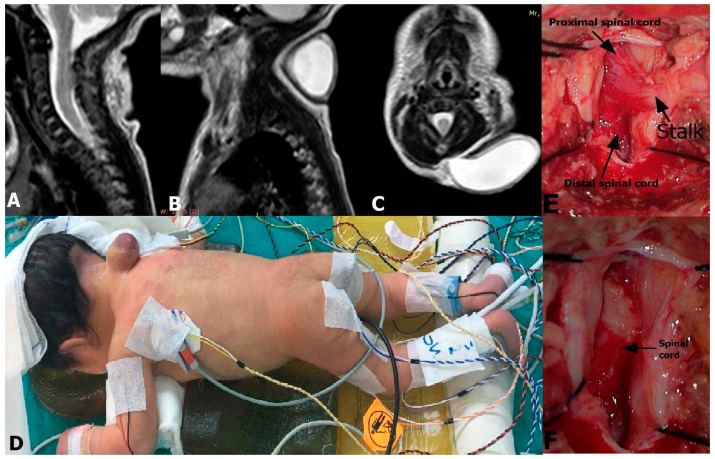
Sagittal (**A**), (**B**) and axial (**C**) T2-weighted magnetic resonance imaging (MRI) show cerebrospinal fluid (CSF)-filled meningocele sac and stalk lying from the cervical spinal cord to the inner wall of the sac. (**D**) The patient underwent surgical treatment under intraoperative neuromonitoring. (**E**) After the removal of the meningocele sac, the stalk and the spinal cord became obvious. (**F**) Spinal cord was detethered after the cutting of stalk at its origin.

**Figure 2 medicina-55-00028-f002:**
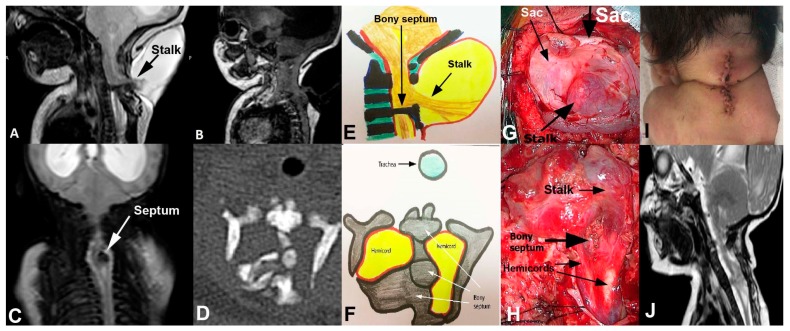
Sagittal T2-(**A**) and T1-weighted (**B**) Magnetic resonance imaging (MRI) show the stalk, cerebrospinal fluid (CSF)-filled meningocele sac and bony septum below the origin of stalk. Coronal T2-weighted MRI (**C**) and axial computed tomography (CT) scan (**D**) show bony septum dividing the spinal cord into 2 hemicords. It is obvious that the septum was fragmented. The line drawings (**E**), (**F**) depict the location of stalk and septum, as well as hemicords. Following the opening of meningocele sac, the stalk became visible (**G**). Then, the bony septum was removed and the dura was opened. The hemicords became obvious before cutting the stalk from its origin (**H**). The skin was closed (**I**) and postoperative 3rd month sagittal T2-weighted MRI show the lack of stalk and bony septum (**J**).

**Figure 3 medicina-55-00028-f003:**
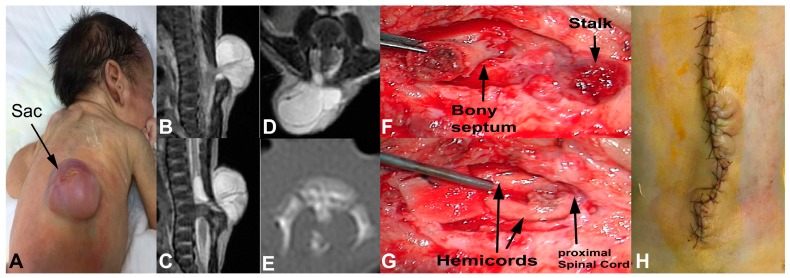
Patient was presented with cystic thoracolumbar lesion (**A**). Sagittal T2-weighted magnetic resonance imaging (MRI) (**B**), (**C**) show the stalk, cerebrospinal fluid (CSF)-filled sac and the bony septum below the stalk level. Coronal T2-weighted MRI (**D**) and axial computed tomography (CT) scan (**E**) show bony septum which divides the spinal cord into the 2 hemicords. After the removal of meningocele sac and laminectomy, the bony septum and the stalk became visible (**F**). The bony septum was removed and the stalk was cut. Proximal spinal cord and both hemicords are now untethered (**G**). The skin was closed (**H**).

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
