# Peer review of "Limited Dorsal Myeloschisis with and without Type I Split Cord Malformation: Report of 3 Cases and Surgical Nuances"

_medicina, 2019, doi:10.3390/medicina55020028_

Reviewer 1 Report

The authors present a very interesting series of 3 patients with limited dorsal myeloschesis. Two of these patients also had Type 1 Split Cord Malformation. While the authors do not present any major insights or innovations related to the treatment of these conditions, they have done a very nice job describing the presentation and the surgical treatment. 

One clarification is needed: In the introduction, they make mention of an association between Chiari 2 malformation and LDM. I am not aware of a linkage between LDM and Chiari 2. If the authors are to make this claim, then more detailed discussion of a reference is necessary.

The authors also note that the embryologic basis for LDM is a mystery. While there may be some debate about this, the work of Pang (that they have cited) makes a reasonably compelling argument for incomplete disjunction between cutaneous and neural ectoderm as the underlying lesion.

In summary, this is an interesting case report and would be a welcome addition to the literature.

Author Response

One clarification is needed: In the introduction, they make mention of an association between Chiari 2 malformation and LDM. I am not aware of a linkage between LDM and Chiari 2. If the authors are to make this claim, then more detailed discussion of a reference is necessary.

Reply: We deleted this statement. This sentence is corrected  as “Frequent associated pathologies are hydrocephalus and split cord malformation (SCM)”.

The authors also note that the embryologic basis for LDM is a mystery. While there may be some debate about this, the work of Pang (that they have cited) makes a reasonably compelling argument for incomplete disjunction between cutaneous and neural ectoderm as the underlying lesion.

Reply: We deleted the sentence “The embryologic basis of this association is a mystery”.

Reviewer 2 Report

This paper adds to an unusual entity. I think the authors operative techniques and conclusions are valid. This is a good reference for the young neurosurgeon who will be seeing complex neural tube patients. The paper has an excellent reference section for those interested in the historical aspects of dealing with unique situations.

I would be a bit more emphatic about the last sentence. Intraoperative monitoring is a necessary requirement in these cases. It adds another layer of safety especially during manipulation of the cord to remove the bone or cartilaginous abnormalities. Since children can be fragile under anesthesia with potential blood pressure fluctuations, monitoring can detect these changes  early and appropriate actions taken  when the surgeon is concentrating on the operative aspects especially while performing micro-dissection.

Author Response

I would be a bit more emphatic about the last sentence. Intraoperative monitoring is a necessary requirement in these cases. It adds another layer of safety especially during manipulation of the cord to remove the bone or cartilaginous abnormalities. Since children can be fragile under anesthesia with potential blood pressure fluctuations, monitoring can detect these changes  early and appropriate actions taken  when the surgeon is concentrating on the operative aspects especially while performing micro-dissection.

Reply: We deleted the last sentence “Intraoperative neuromonitoring may be useful to check the functionality of spinal cord, hemicords, and aberrant rootlets that penetrate into the meningocele sac”.

Round  2

Reviewer 1 Report

The authors have addressed my concerns. 

Author Response

We corrected the manuscript for English language and style.

Reviewer 2 Report

I would not remove the sentence about monitoring I think it is important. I would just change it to say that monitoring adds  to the safety of the procedure, or something like that. your statement was monitoring 'may' ...There is no question monitoring is useful and in some places would be standard of care.

Author Response

I would not remove the sentence about monitoring I think it is important. I would just change it to say that monitoring adds  to the safety of the procedure, or something like that. your statement was monitoring 'may' ...There is no question monitoring is useful and in some places would be standard of care.

Our reply: We corrected this sentence as "Intraoperative neuromonitoring is useful to check the functionality of spinal cord, hemicords, and aberrant rootlets that penetrate into the meningocele sac and provides safe surgery for children."

We also corrected the manuscript for English language and style.